# Modulation of Innate Immune Toxicity by Silver Nanoparticle Exposure and the Preventive Effects of Pterostilbene

**DOI:** 10.3390/ijms22052536

**Published:** 2021-03-03

**Authors:** Rong-Jane Chen, Chiao-Ching Huang, Rosita Pranata, Yu-Hsuan Lee, Yu-Ying Chen, Yuan-Hua Wu, Ying-Jan Wang

**Affiliations:** 1Department of Food Safety/Hygiene and Risk Management, College of Medicine, National Cheng Kung University, Tainan 70101, Taiwan; janekhc@gmail.com (R.-J.C.); winds031215@gmail.com (C.-C.H.); rositaphan@gmail.com (R.P.); 2Department of Cosmeceutics, China Medical University, Taichung 40402, Taiwan; yhlee@mail.cmu.edu.tw; 3Department of Environmental and Occupational Health, College of Medicine, National Cheng Kung University, Tainan 70101, Taiwan; 101312123@gms.tcu.edu.tw; 4Department of Radiation Oncology, National Cheng Kung University Hospital, National Cheng Kung University, Tainan 704, Taiwan; 5Department of Medical Research, China Medical University Hospital, China Medical University, Taichung 40402, Taiwan

**Keywords:** silver nanoparticles, zebrafish, innate immune toxicity, pterostilbene, cytokines

## Abstract

Silver nanoparticles pose a potential risk to ecosystems and living organisms due to their widespread use in various fields and subsequent gradual release into the environment. Only a few studies have investigated the effects of silver nanoparticles (AgNPs) toxicity on immunological functions. Furthermore, these toxic effects have not been fully explored. Recent studies have indicated that zebrafish are considered a good alternative model for testing toxicity and for evaluating immunological toxicity. Therefore, the purpose of this study was to investigate the toxicity effects of AgNPs on innate immunity using a zebrafish model and to investigate whether the natural compound pterostilbene (PTE) could provide protection against AgNPs-induced immunotoxicity. Wild type and neutrophil- and macrophage-transgenic zebrafish lines were used in the experiments. The results indicated that the exposure to AgNPs induced toxic effects including death, malformation and the innate immune toxicity of zebrafish. In addition, AgNPs affect the number and function of neutrophils and macrophages. The expression of immune-related cytokines and chemokines was also affected. Notably, the addition of PTE could activate immune cells and promote their accumulation in injured areas in zebrafish, thereby reducing the damage caused by AgNPs. In conclusion, AgNPs may induce innate immune toxicity and PTE could ameliorate this toxicity.

## 1. Introduction

Silver nanoparticles (AgNPs) are widely used in various medical and consumer products such as cosmetics, textiles, antibacterial agents, personal care products and medical devices [1]. The global usage of AgNPs has grown exponentially, thereby increasing the amount of AgNPs that enter the environmental compartments including plants, animals, and humans, through a variety of routes [1]. Due to their small size, AgNPs can pass through biological systems and enter cells to trigger undesired and hazardous effects [2]. The toxicity of AgNPs is related to their characters, such as their shape, size, concentration, chemical coating, surface charge, aggregation and synthesis processes [3]. After exposure, AgNPs accumulate and induce toxic effects on the immune system, neurons, kidney, liver, lung, gastrointestinal tract, and reproductive system [4]. The modulation of the immune response by AgNPs is a unique area of interest because uncontrolled immune stimulation or immune suppression could lead to allergic reactions or reduce the body’s immune response to damage, infection, and cancers [5]. Currently, AgNPs have been reported to alter the morphology and viability of freshly isolated human peripheral blood mononuclear cells (hPBMCs) [6]. Moreover, pregnant mice exposed to AgNPs exhibited immunological dysfunction in the dam and placenta [7]. These disturbing effects on the immune response indicated the possible AgNPs immunotoxicity. At present, the role of AgNPs in modulating the immune response is extremely novel, and the mechanisms underlying their toxicity are still under investigation.

Currently, a large amount of research conducted on immunotoxicity has focused on ex vivo or in vitro experiments using cell lines from various human organs, such as A549 and Caco-2 cells. However, in vitro experiments lack the complex physiology of tissue barriers or the innate immune system provided by the in vivo models [8]. As there is no possibility of studying the toxicity of NPs in humans and considering the 3R (refine, reduction, and replace) principle of animal studies, alternative vertebrate models, such as zebrafish [9] have been developed [10]. Embryonic and adult zebrafish have been used as ideal alternative biological models for testing toxicity and for screening environmental toxicants, chemicals, drugs, and food additives; zebrafish have also been used by regulatory agencies for chemical safety assessments [11]. Zebrafish are also a good model for addressing questions regarding immunity [11]. During the initial 4 days, the innate immune system develops in zebrafish but has no adaptive immunity markers [11]. After 4 days, adaptive immunity develops, but it takes 4–6 weeks to develop a fully functional adaptive immune system. Macrophages, neutrophils, and dendritic cells are indispensable for innate immunity in vertebrates and play similar roles in zebrafish. The functional properties of the immune effector proteins, including Toll-like receptors (TLRs), NFκB, cytokines and chemokines, can also be evaluated [11]. Accordingly, in the early development period of zebrafish, it is possible to exclusively study the innate immune response. Therefore, in the present study, we used zebrafish embryos to evaluate the innate immune response regulated by AgNPs.

AgNPs are the most frequently produced and marketed metal NPs and by far the most commonly used NPs in medical and consumer products [2]. However, their toxicity effects should be cause for concern. Nonspecific oxidative stress has been suggested as one of the greatest concerns of AgNPs-induced toxicity. The production of reactive oxygen species (ROS) leads to numerous cytotoxicities, such as programmed cell death, genotoxicity, and alteration of immune function [12]. Hence, reduction of this oxidative stress could offer protection against AgNPs-induced immunotoxicity, and make the use of AgNPs safer. In this regard, pterostilbene (PTE, trans-3,5-dimethoxy-4-hydroxystilbene), a naturally occurring analog of resveratrol and is a stilbene compound with many health benefits, was used in the study. The protective effects of PTE include its antioxidant activity, anti-inflammatory effects, anti-obesity, and chemopreventive effects, which are attributed to its unique structure [13,14,15]. Current studies also indicated the modulation effects of PTE on the immune response. For instance, PTE inhibited ROS production in neutrophils and increased neutrophil numbers in a model of arthritis [16]. Furthermore, PTE has the potential to enhance the innate immune response, particularly in combination with 1α,25(H)2D3 [17]. In our study, we encapsulated the PTE with poly(lactic-co-glycolic acid) (PLGA) called PLGA-PTE, which is a biodegradable, synthetic polymer that fulfills the requirements of high performance and safety and thus will not induce inflammation by itself. Consequently, our present study aimed to evaluate the immunotoxicity of AgNPs and to investigate the protective effects of PLGA-PTE.

## 2. Results

### 2.1. Characterization of AgNPs and PLGA-PTE

The full characteristics of AgNPs are presented in Figure 1 and Table 1. TEM images showed that the synthesized citrate-capped AgNPs exhibited a spherical-shape and good dispersibility (Figure 1A). The average diameter of the AgNPs was 8.89 ± 1.68 nm (Table 1). The hydrodynamic size (Figure 1B,C and Table 1) of the AgNPs in ultrapure water was 13.2 ± 3.1 nm. The resulting zeta potential of the AgNPs was approximately −28.85 mV (Table 1). The UV-Vis analysis showed an absorption peak of Ag at 391 nm, along with the results of energy dispersive X-ray analysis (EDX), which showed that the purity of AgNPs was higher (Figure 1E).

The characteristics of PLGA-PTE was shown in Figure 2. The results of HPLC confirmed that PTE was incorporated into PLGA (Figure 2A) The presented size of PLGA-PTE was 56.9 ± 13.6 nm (Figure 2B). Characterization by TEM showed that PLGA-PTE had a spherical morphology (Figure 2C). The results of UV-Vis analysis showed an absorption peak of PTE at 310 nm and a peak of PLGA at 190 nm (Figure 2D,E). The PLGA-PTE absorption peaks were identified at 310 and 190 nm respectively, confirming that the PLGA-PTE was successfully synthesized (Figure 2F). In addition, the release rate of PTE from PLGA in water was detected by UV-Vis spectrophotometer showed that PLGA particles provided a sustained release of PTE of linear character, following an initial burst period (releasing approximately 30% within the first day). The release rate of PLGA-PTE in water is approximately 30–50% within 120 h (Figure 2G).

### 2.2. Effects of AgNPs and PLGA-PTE on Zebrafish Embryos

Zebrafish embryos were exposed to AgNPs (0.1, 0.2, 0.3, 0.4, and 0.5 μg/mL) from 0–120 hpf (hours post fertilization) (Figure 3A). The results showed that AgNPs reduced the survival of the embryos in a dose- and time-dependent manner. In contrast, pretreatment with 0.3 μg/mL PLGA-PTE followed by 0.3 μg/mL AgNPs for 0–120 hpf significantly reversed the mortality of the embryos (Figure 3B). Figure 3C further shows that embryos exposed to AgNPs had higher aggregation of NPs in the chorion, whereas PLGA-PTE reduced the aggregation of AgNPs. Remarkably, AgNPs induced abnormalities (Figure 4A) and malformations during embryonic development (Figure 4B), including pericardial edema (PE), yolk sac edema (YSE) [19], axial curvature (AC), and tail malformation (TM) (Figure 4B). We observed that the exposure of AgNPs affected the function of early cell proliferation in the caudal fins, while pretreatment with PLGA-PTE reversed the malformation response (Figure 4C).

### 2.3. Innate Immune Response Affected by Exposure to AgNPs and PLGA-PTE Alone or Combined

To assess whether the innate immune response was affected, zebrafish transgenic lines Tg (mpx: eGFP): Neutrophiles and Tg(cd41:eGFP//mpeg1:mCherry): Macrophages were observed in the study using green and red fluorescence, respectively. All groups of zebrafish were exposed to the treatment every day, and neutrophil and macrophage fluorescence images of 3 dpf (days post fertilization) and 5 dpf zebrafish embryos were captured. In vivo imaging of the zebrafish embryos showed that neutrophil and macrophage fluorescence in the AgNPs groups was relatively high at 3 dpf; however, the fluorescence was decreased after 5 dpf. Interestingly, the numbers of neutrophils and macrophages were increased in the PLGA-PTE pretreatment groups at both time points, indicating the potential of PLGA-PTE to protect against AgNPs-induced immunotoxicity (Figure 5A,B). Next, R6G-AgNPs were used to determine the content of AgNPs in zebrafish embryos. The results showed the images of R6G-AgNPs accumulation, confirming the internalization of AgNPs in zebrafish (Figure 6A). In addition, the fluorescence of neutrophils after R6G-AgNPs treatment exhibited a similar trend as after treatment with AgNPs without the R6G dye, as previously described in Figure 5.

To investigate the correlation between the immune response and oxidative stress with the protective effect of PLGA-PTE, the generation of ROS was measured by DCFH-DA. The results indicated that the AgNPs treatment groups showed stronger fluorescence than the other groups, indicating that exposure to AgNPs increased ROS production (Figure 6C). In contrast, pretreatment with PLGA-PTE significantly reduced ROS production, suggesting the antioxidative stress effects of PTE (Figure 6C). Previous studies have indicated that mitochondria are likely to be the intracellular target of AgNPs toxicity [20]. AgNPs induced brain toxicity has been associated with abnormal mitochondrial function and the upregulation of the genes relevant to innate immunity [21]. Therefore, we further explored whether PLGA-PTE restores the function of immune cells by regulating mitochondria damage. Human macrophage cell line THP-1 cells were treated with AgNPs for indicated time points and the results showed that AgNPs significantly reduced mitochondrial membrane potential (MMP) (Figure 6E,F) and induced ROS production (Figure 6G,H) as evidenced by MitoTracker™ Red and MitoSOX™ staining, respectively. Meanwhile, pretreatment with PLGA-PTE for 1 h prior to AgNPs treatment has significantly protected against MMP loss and ROS production in macrophages (Figure 6E–H). These findings suggested that mitochondria are a target for AgNPs toxicity, while PLGA-PTE protected against mitochondria damage, and thereafter reversed MMP and reduced ROS production. The in vitro results were consistent with the findings observed in zebrafish.

### 2.4. The Addition of PLGA-PTE Ameliorates the Toxic Effect of AgNPs in Immune Cells

A tail-transection test was performed to further understand whether the immune function was altered after exposure to AgNPs. After the transgenic zebrafish were anesthetized, their tail fins were transected at the end of the spinal cord by a sterile blade under a microscope. The reverse migration of the immune cells was observed under a fluorescence microscope after 1 h. The results showed that neutrophil-eGFP fluorescence accumulated faster than macrophage-mCherry fluorescence at the injured site in the control groups (Figure 7A,C,D). After 2 h, the reverse migration of neutrophils and macrophages was detected toward the injured tail fin in the control groups (Figure 7B–D). However, the reverse migration of neutrophils and macrophages was reduced in the AgNPs treatment groups, indicating that the functions of neutrophils and macrophages were affected (Figure 7). Substantially, PLGA-PTE pretreatment increased the migration of neutrophils and macrophages to the injured sites, suggesting that PLGA-PTE protected the immune response in zebrafish (Figure 7C,D).

To explore the immune regulatory pathways affected by AgNPs and PLGA-PTE in zebrafish embryos, RT-qPCR was used to analyze the expression of related genes. The RT-qPCR results showed that the expression of Toll-like receptors 2 (*TLR2*), *IL-1β*, *IL-8*, *IL-10*, *IL-26*, Lysozyme (which can catalyze the hydrolysis of the bacterial mucopolysaccharide cell wall), *Myd88* (which is responsible for the detection of xenobiotics on the cell membrane), and *NF-κB* was downregulated by AgNP treatment, while PLGA-PTE pretreatment restored the expression of these genes (Figure 8). In contrast, the expression of *IL-6* and *IL-22* showed slight upregulation in the AgNPs treatment groups in a dose-dependent manner. The results indicated that the immune response could be affected by AgNPs, but PLGA-PTE could provide protective effects on the immune response.

## 3. Discussion

Our present study aimed to investigate the toxicity of the innate immune system induced by exposure to AgNPs and further analyze whether the addition of PLGA-PTE could ameliorate this toxicity. We used citrate-capped AgNPs and PLGA-PTE that was synthesized in-house. In the present study, we exposed wild-type zebrafish embryos to AgNPs to determine the LC_50_, which was used as the concentration in the subsequent study. The results showed that AgNPs significantly induced innate immune toxicity, which was observed in the tail transection test and the reverse migration of the immunocytes. Moreover, ROS production and alterations in gene expression levels related to innate immune function were detected after exposure to AgNPs. Significantly, the addition of PLGA-PTE prevented toxicity to zebrafish embryos, reduced ROS production, and prevented innate immune toxicity compared to AgNPs alone. Taken together, our study is the first to indicate that AgNPs could induce innate immune toxicity in zebrafish, whereas PLGA-PTE significantly prevents innate immune toxicity.

For dose-range testing, we exposed zebrafish embryos to AgNPs, and toxic effects were induced during zebrafish embryonic development. Morphological abnormalities (Figure 3) and embryo malformations (Figure 4), namely pericardial edema (PE), yolk sac edema (YSE) [19], axial curvature (AC), and tail malformation (TM) were observed. Similar to our study, other studies have also reported corresponding observations in zebrafish embryos following exposure to AgNPs [22,23]. Additionally, our study demonstrated that the exposure of AgNPs affected early cell proliferation in the caudal fins, which was similar to the result of other studies [24,25]. Consistent with our findings, a previous study showed that AgNPs induced embryonic developmental toxicity and cardiovascular toxicity in zebrafish [26]. These observations also provide evidence to establish safety limits for AgNPs exposure to the external environment. To prevent these toxicities, our results indicated that pretreatment with PLGA-PTE attenuated malformation and death in zebrafish, as shown in Figure 4C.

After the physiological and/or anatomical barrier is breached, the innate immune system is the next barrier to prevent foreign particles from entering the human body. The innate immune system drives the adaptive immune system to carry out subsequent responses. Accordingly, the innate immune system can directly react with penetrating harmful substances and induce inflammation. Although there are currently no reports regarding NPs-induced immune disease, however, exposure to NPs may interfere with various cell activities, including modification of membrane receptors, gene expression, and cytokine production [27]. Such effects have been reported as the underlying mechanism of many different organ diseases. In addition, chemicals that affect natural killer cells [8], natural killer T cells (NKT), macrophages, CD8+ cell cytotoxicity, and T lymphocytes or alter the production of cytokines, may impair the immune system against infections [28]. Specifically, for AgNPs, considerable research has shown that exposure to these recurring metallic nanoparticles may affect the immune system, either directly or indirectly. For instance, exposure to AgNPs may trigger neutrophils to induce inflammation and secrete chemokines [29]. Furthermore, once internalized, AgNPs also release Ag^+^ ions and promote ROS generation [10]. Herein, our results are in accordance with the previous studies showing that AgNPs induced the production of ROS and altered the function of neutrophils and macrophages, which are two of the most important cells in innate immune function.

Multiple publications have proven that the physicochemical properties of NPs are one of the key factors that affect the innate immune response [8]. Differences in size, distribution, surface charge, surface area, reactivity, crystallinity, aggregation in certain solvents, composition, surface coating, synthesis methods, and impurities affect not only the biodistribution and cell absorption of nanoparticles but also the innate immune response. Among these characteristics, the surface modification of nanoparticles is the most relevant to the immune system, and the outer coating determines the toxicity of nanoparticles [30]. In comparison, another study exposed primary human blood mononuclear cells to AgNPs with particle sizes of 5 nm and 28 nm to study the innate immunological effects. The results showed that smaller particles have a higher potential to activate innate immunity, which was measured by the production of IL-1β and the induction of inflammatory body formation and other effects [31]. Moreover, smaller particles also induce greater cytotoxicity in monocytes and macrophages [32]. Similarly, in our study, the average particle size of AgNPs used in this experiment was 8.89 ± 1.68 nm, and the surface had been modified using citrate, which might have a higher possibility of stimulating the innate immune response.

Oxidative stress is the main mechanism of nanoparticle-induced toxicity [33]. The results of our study further showed that the oxidative stress detected in the AgNPs-treated group was higher than that in the control group (Figure 6). Similarly, the reverse migration ability of the immune cells in the AgNPs-treated group in the tail-cutting test was reduced (Figure 7), indicating that the function of immune cells was also affected. In our study, the results further showed that most of the AgNPs stained with R6G fluorescent dye accumulated near the gastrointestinal (GI) tract of the embryos (Figure 6). Our results further confirmed that ROS production could also be detected using THP-1 cells which are commonly used immune cells in studies, and further indicated that mitochondria were damaged by AgNPs. In contrast, PLGA-PTE significantly protected MMP loss and reduced ROS production that could partially explain the protective role of PTE in reducing innate immunity caused by AgNPs in zebrafish. In addition to the role of mitochondria damage, the intake of AgNPs could interfere with the uniformity of intestinal microbes and the diversity of the flora in a dose-dependent manner, indicating that AgNPs might also cause toxicity to the intestinal microbiota [18]. The relationship between the intestinal microbiota and the immune system is inseparable in that the regulation of intestinal bacteria relative to the immune system and the metabolism of the host are very important [34]. In addition to the intestinal flora affecting the local immune response, this microbiota also has a broader role, namely, to maintain the balance of innate and adaptive immunity [19]. In Figure 5 and Figure 6, the experimental results showed that at 5 dpf, the AgNPs-treated group had a significantly lower number of neutrophils and macrophages distributed around the intestines than the other groups. However, the relationship of the zebrafish gut microbiota with the immune response was not explored in our study.

Interactions between nanoparticles and the innate immune system can affect the adaptive immune response through the production of cytokines and chemokines. Another publication previously stated that the IL-1β produced by monocytes in response to exposure to AgNPs is a key cytokine that is involved in lymphocyte activation and proliferation [32]. In our study, exposure to AgNPs also affected the expression of inflammatory immune-related cytokines and chemokines, such as *IL-1β*, *IL-10*, and *IL-8*. Figure 8 shows that along with the increase in the concentration of AgNPs, the expression of the *TLR2*, *Myd88*, *IL-8*, *NF-κB*, and *IL-1β* genes was downregulated and the addition of PTE could restore the production of each protein. IL-1β is mainly produced by macrophages, monocytes, fibroblasts, and dendritic cells, which play important roles in the organism’s fight against infection [35]. The recruitment of innate immune cells is a characteristic of the inflammatory response. IL-1β can facilitate the transfer of immunocompetent cells to the infected area in zebrafish [36]. The present results suggested that reduced IL-1β could be associated with the impairment of the innate immune response caused by AgNPs. Previous studies have stated that AgNPs can activate various pathways, including the MAPK and NF-κB pathways, leading to the transcription of many genes involved in inflammatory responses and triggering harmful inflammatory responses [37]. However, a low dose of AgNPs slightly upregulated the expression of *NF-κB* and *IL-1β*, and these levels were downregulated with increasing doses. Furthermore, for all genes, the groups that were exposed to PTE alone showed no significant changes compared to the control groups (Figure 8). This outcome indicated that exposure to AgNPs interferes with the regulation of immune-related genes, and the addition of PTE helps maintain the balance of the immune system. Among these genes, *IL-8* is a chemokine of neutrophils that can induce neutrophils to accumulate at the site of infection. This result corresponds to the results of previous experiments. From the observations of the embryo tail 1 h post cutting presented in Figure 7A, the number of neutrophils that accumulated at the wound site in the AgNPs-treated group was significantly lower than that of the other groups. Since neutrophils are the first-line cells that respond to injury, macrophages can be recruited to the inflamed tissue to engulf pathogens and tissue debris [38]. The results indicated that AgNPs impaired the normal function of innate immune cells.

As the world keeps searching for alternative natural compounds, the translational values of stilbene compounds are hard to ignore. It has been proven in many publications that PTE, one of the stilbene compounds, is beneficial on various levels and applications [39]. Notably, our study shows that pretreatment with PTE increased the survival rate of embryos exposed to AgNPs and reduced the occurrence of malformations (Figure 3 and Figure 4). Compared with the group that was treated with AgNPs alone, the combination with PTE activated immune cells and promoted immune cells to accumulate in the damaged areas in the PLGA-PTE + AgNPs-treated group (Figure 7). Other results also show that in the in vitro model, the addition of PTE can inhibit the production of active oxides of neutrophils and effectively regulate the number of neutrophils in the in vivo arthritis test [16]. In our previous publication, we summarized the superior characteristics and various applications of this compound [16,40]. PTE has been reported to suppress the expression of IL-1β in the RAW264.7 inflammation model due to its anti-NF-κB property [41]. The ROS scavenging properties of PTE are aligned with the findings of our present study (Figure 6C), indicating that the health benefits of PTE could be attributed to its modulation of inflammatory responses.

Taken together, the findings reported in our present study have confirmed that exposure to AgNPs modulates the innate immune system of zebrafish, and we performed RT-qPCR to further investigate the upregulation and downregulation of genes that lead to inflammation. AgNPs are the most widely used NPs in consumer products. However, their toxicity has raised concerns, limiting their usage. In this context, our study presents an answer to ameliorate the immunotoxicity induced by AgNPs. Pretreatment with PLGA-PTE could reduce the toxic effects of AgNPs in zebrafish embryos. PTE is a protective compound with various superior characteristics, namely, anticancer and antioxidant properties. Previously, we gathered evidence on how PTE can suppress oxidative stress and inflammation, and modulates senescence, apoptosis, autophagy, and cell cycle arrest [14]. In addition, PLGA-capped PTE can increase the solubility of PTE in water (Figure 2) and change the characterization of PTE to provide better bioavailability in zebrafish. However, further investigation of how PLGA-PTE protected against innate immune toxicity after nanoparticle exposure is still needed. In summary, the addition of PTE is a promising solution to ameliorate the immune toxicity induced by AgNPs, thereby providing a safer approach for its application.

## 4. Materials and Methods

### 4.1. Synthesis of AgNPs and PLGA-PTE

For AgNPs synthesis, 50 mL of 20 mM silver nitrate (AgNO_3_) and 40 mL of 80 mM sodium citrate were mixed. Subsequently, 890 mL of distilled water, 50 mL of 20 mM nitric acid, and 20 mL of 100 mM sodium borohydride (NaBH_4_) were added and stirred for 2 h. The AgNPs solution was centrifuged at 7500 RCF for 30 min to collect the small particles in the upper layer, which were further centrifuged at 12,500 RCF for 2 h. The lower layer of the precipitated AgNPs was collected and stored at 4 °C. Atomic absorption spectroscopy [42] (PerkinElmer AAnalyst™ 600, PerkinElmer, Inc., Shelton, CT, USA) was utilized to determine the concentration.

For PLGA-PTE synthesis, 50 mg of PLGA and 10 mg PTE were dissolved in 5 mL acetone. EtOH:H_2_O = 1:1 solution was added dropwise into the mixed solution until it was completely dissolved, and stirring was continued until the solution became turbid. Then, the mixture was stably injected using a syringe into 20 mL of distilled water and stirred for 10 min. The organic solvent was vacuumed. The resulting supernatant was discarded and dissolved by adding water. PLGA-PTE was diluted with acetonitrile [43] to yield 10, 25, 100, 200, and 500 ppb of standard solutions and stored at 4 °C. The concentration of the synthesized PLGA-capped PTE (PLGA-PTE) was determined using high-performance liquid chromatography (HPLC) (Waters 600E, Waters Co., Milford, MA, USA). The detector used was Waters Nova-Pak C-18 column (150  ×  3.9 mm, 5 μm particle size). The mobile phase is a 50:50 vol./vol. mixture that was filtered and degassed under reduced pressure.

### 4.2. AgNPs and PLGA-PTE Characterization

Following the syntheses, the AgNPs and PLGA-PTE were characterized. For stability testing, an ultraviolet-visible (UV-Vis) spectrophotometer (Thermo Scientific™ NanoDrop 2000c, NanoDrop Technologies, Thermo Fisher Scientific, Inc., Pittsburgh, PA, USA) was used. The UV spectra ranged from 200 to 800 nm. Images of NP morphology were obtained using transmission electron microscopy [18] (JEOL Co., Akishima, Tokyo, Japan). The energy-dispersive X-ray spectroscopy (EDX) method was performed to obtain elemental composition analysis using an EDX spectroscope (JEM-2100F Field Emission Electron Microscope [200 kV]; JEOL Ltd., Tokyo, Japan). Subsequently, a built-in scanning transmission electron microscope (STEM) (JEOL ARM200F, JEOL Ltd., Tokyo, Japan) was utilized to analyze the constituent elements.

The hydrodynamic diameter was measured with a dynamic light scattering (DLS) instrument (Delsa™ Nano C, Beckman Coulter, Inc., Brea, CA, USA). Using the same instrument, phase analysis light scattering (PALS) was performed to obtain the zeta potential and polydispersity index (PI) of the samples.

### 4.3. Drug Loading and Drug Release Analysis of PLGA-PTE

For the drug loading study of PLGA-PTE, 500 µl of PLGA-PTE solution was centrifuged at 21,100 RCF for 20 min to separate the PLGA-PTE nanoparticles and free PTE. The PLGA-PTE NPs pellet was then redissolved in alcohol and the PTE concentration was analyzed using ultraviolet-visible (UV-Vis) spectrophotometer. The UV spectra ranged from 200 to 800 nm. The maximum absorbance value was detected at 310 nm. The drug loading rate (%) was calculated by using the following formula: (Amount of drug released from the PLGA-PTE/Initial amount of drug for NPs preparation) × 100%. All of the experiments were done in triplicate. The loading rate is around 49.00 ± 1.36 (%).

The release of PLGA-PTE was carried out in distilled water. First, PLGA-PTE was centrifuged at 21,100 RCF for 20 min to separate After 6, 24, 48, 72, 96, and 120 h, the solution was withdrawn and centrifuged at 21,100 RCF for 20 min to separate the nanoparticles and the free PTE. The release of PTE was analyzed with UV-visible spectrophotometry at 310 nm. All of the experiments were done in triplicate.

### 4.4. Zebrafish Maintenance and Treatment

Zebrafish embryos were cultured and collected from wild-type and the transgenic zebrafish Tg(mpx:eGFP):Neutrophils and Tg(mpeg1:mCherry):Macrophages. Both types of transgenic zebrafish were sourced from the National Health Research Institutes (NHRI), Taipei, Taiwan. All the experiments on the zebrafish embryos were performed in accordance with the latest version of the OECD test guidelines (TG 236: Fish Embryo Acute Toxicity (FET) Test) and the Guide for Care and Use of Laboratory Animals, National Cheng Kung University, College of Medicine. All the experiments were approved by the Institutional Animal Care and Use Committee of National Cheng Kung University, Taiwan (approval no. 107204, 10 May 2018). The swimming tanks for the adult zebrafish were maintained at temperatures ranging from 28 °C ± 1 °C with a light/dark cycle of 14-h/10-h cycle to ensure the natural circadian rhythm. Zebrafish are fed daily with brine shrimp. A total water quality test was performed once every 2 weeks.

For the acute toxicity test, zebrafish embryos were deposited in 12-well plates and treated beginning at 3 to 4 h post-fertilization (hpf) for five days (120 h). In each well, 10 zebrafish embryos were deposited with three biological replicates. The AgNPs concentrations tested were 0, 0.1, 0.2, 0.3, 0.4, and 0.5 μg/mL to identify the nonlethal doses to investigate their effects on the immune system. The AgNPs solution was replaced every 24 h, and the morphology and survival rate were observed and recorded at 24, 48, 72, 96, and 120 hpf using a Nikon SMZ-800 stereomicroscope equipped with an MD-500 Digital Microscope Camera (Nikon Corporation, Tokyo, Japan) for storage. At the end of the 120 hpf experiment, the LC_50_ was found to be nearly 0.3 μg/mL; hence, this dose was selected for subsequent studies. In addition, zebrafish embryos were exposed to 0, 0.3, 0.5, and 1 μg/mL of PLGA-PTE aqueous solution and observed at 24, 48, 72, 96, and 120 hpf. At the end of the 120 hpf experiment, the embryo survival rate at the treatment of 0.3 μg/mL PLGA-PTE was found to be 90%; hence, this dose was selected for subsequent studies. To further prove the protective effects of PLGA-PTE addition to AgNPs, the zebrafish embryos were divided into 4 groups: (1) control, (2) 0.3 μg/mL PLGA-PTE, Ref. [9] PLGA-PTE + AgNPs (embryos were pretreated with PLGA-PTE 0.3 μg/mL, followed by exposure to 0.3 μg/mL), and [9] 0.3 μg/mL AgNPs.

### 4.5. Functional Analysis of Immunocytes

The immune cell number and AgNPs aggregation inside the exposed zebrafish larvae were observed under a vertical fluorescence microscope (BX51, Olympus Co., Tokyo, Japan). To understand the function of the immune cells after exposure to AgNPs, a tail transection test was performed. After 120 h of exposure, the transgenic zebrafish larvae were anesthetized. Thereafter, the caudal fins were transected using a sterilized scalpel, and the fish were returned to their original group. At 1 h and 2 h after the second exposure, the zebrafish were anesthetized once again, and the reverse migration of the immunocytes was observed under a vertical fluorescence microscope [44].

### 4.6. Reactive Oxygen Species (ROS) Analysis

Prior to treatment of the zebrafish embryos, the AgNPs were stained using 2′,7′-dichlorofluorescein diacetate (DCFH-DA) dye. At the end of the 120 hpf experiment, the ROS production in the zebrafish embryos in response to exposure to AgNPs was observed using a vertical fluorescence microscope.

### 4.7. Mitochondrial Damage Analysis

To analyze the mitochondrial damage induced by exposure to AgNPs, mitochondrial membrane potential (MMP) analysis and mitochondrial superoxide analysis were performed by using THP-1 cells. For MMP analysis, THP-1 cells were exposed to 1 µM PLGA-PTE for 1 h prior to exposure to 6 µg/mL AgNPs. After 6 h of exposure, the level of MMP was measured using the staining of MitoTracker^®^ Red CMXRos probe (M5712, Invitrogen, Rockford, IL, USA). Cells were treated with 100 nM MitoTracker^®^ Red CMXRos and incubated at 37 °C for 30 min and the nuclei were visualized by Hoechst 33258. Images were obtained with Nikon H600L microscope (Nikon, Japan) and the relative fluorescent intensity was measured by ImageJ software.

For mitochondrial superoxide analysis, THP-1 cells were exposed to 1 µM PLGA-PTE for 1 h prior to exposure to 6 µg/mL AgNPs. After 1 h of exposure, the level of mitochondrial superoxide was measured using MitoSOX™ Red probe (MM36008, Invitrogen, Rockford, IL, USA). The images were obtained with Nikon H600L microscope (Nikon, Japan) and the relative fluorescent intensity was measured by ImageJ software (https://imagej.nih.gov/ij/download.html).

### 4.8. Quantitative Reverse Transcription PCR (RT-qPCR)

Quantitative reverse transcription PCR (RT-qPCR) was performed to examine the immune regulatory gene expression level in the zebrafish embryos after exposure to AgNPs and PLGA-PTE treatment. The primers used for the analysis are presented in Table 2.

For quantitative analysis, primers were mixed with SYBR green, deionized water, and cDNA. Further analysis was conducted using the StepOne Plus Real-Time PCR system (Applied Biosystems by Life Technologies, Thermo Fisher Scientific, Inc., Pittsburgh, PA, USA) for 1.5 h.

### 4.9. Statistical Analysis

At least three biological replicates were performed for each experiment. Two-tailed Student’s t-test and ANOVA were used for statistical analysis. Statistical results with a *p*-value < 0.05 were considered significant, and the resulting data were averaged as ±standard deviation (SD).

## 5. Conclusions

The ubiquity of silver nanoparticles (AgNPs) in various consumer products has raised concern for its safety. Multiple publications have proven that exposure to AgNPs will induce toxic effects in the human body. In this study, we show the effects of AgNPs toxicity on immunological functions by using a zebrafish model. Exposure to AgNPs has induced innate immune toxicity, morphological malformations, as well as death in zebrafish embryos. Interestingly, we prove that the addition of pterostilbene (PTE), a naturally recurring stilbene compound, could provide protection against the AgNPs-induced immunotoxicity. We found a significant decrease in the toxic effects found in the groups of zebrafish that were pretreated by PLGA-PTE compared to those without PLGA-PTE pretreatment. Due to the anti-inflammatory and antioxidant properties of PTE, previous publications have stated that PTE was found to inhibit the production of ROS in in vivo arthritis model [16,45]. Because PTE is a lipophilic compound, we encapsulated PTE with PLGA to increase its water solubility, thereby increasing the rate of PTE consumption by zebrafish. The release rate of PTE within 24 h in water is only about 30%, indicating that zebrafish could have consumed enough amount of PLGA-PTE and thereafter induce the protective effects against AgNPs toxicity. Accordingly, our study has proven that the addition of PLGA-PTE can provide a safer application of AgNPs and restore the AgNPs-induced innate immune toxicity. The protective mechanisms of PLGA-PTE in the zebrafish immune system could be partially explained by mitochondria protection and reducing ROS production. Although the detailed protective mechanisms warrant additional investigation, we hope that this study can initiate future studies regarding how PTE can be used to reduce the toxic effects of AgNPs in various aspects.

## Figures and Tables

**Figure 1 ijms-22-02536-f001:**
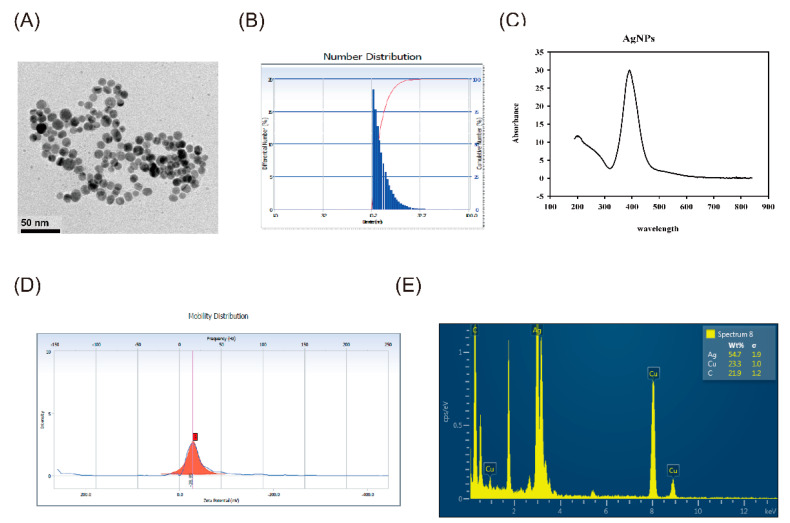
Characterization of Silver nanoparticles (AgNPs). (**A**) Transmission electron microscopy [18] analysis revealed that the synthesized citrate-capped AgNPs were all spherical in shape (scale bar = 50 nm). (**B**) The mean hydrodynamic size of the AgNPs determined by dynamic light scattering (DLS) is 13.2 ± 3.1 nm. The mean size of AgNPs was 8.89 ± 1.68 nm. (**C**) AgNPs were characterized by UV-visible absorption spectra and showed maximum absorbance at 391 nm. (**D**) The zeta potential was observed as a sharp peak at −28.85 mV. (**E**) The energy dispersive X-ray (EDX) spectrum showed that the main components of the NPs are silver.

**Figure 2 ijms-22-02536-f002:**
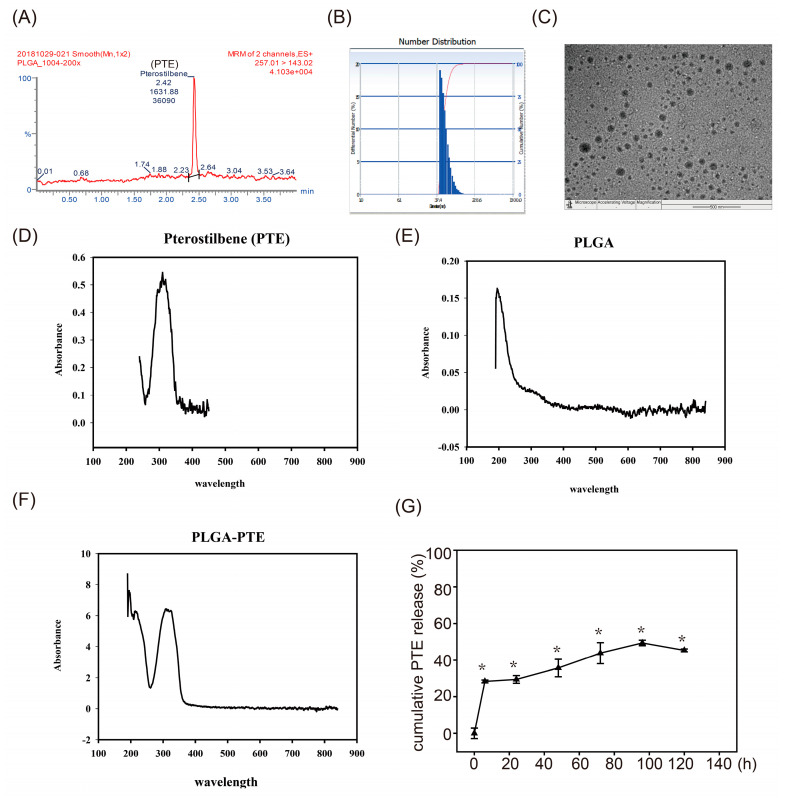
Physicochemical properties of poly(lactic-co-glycolic acid)-pterostilbene (PLGA-PTE). (**A**) HPLC chromatogram of PLGA-PTE (**B**) Transmission electron microscopy analysis revealed that all of the synthesized PLGA-PTE was spherical in shape (scale bar = 500 nm). (**C**) The mean hydrodynamic size of PLGA-PTE determined by dynamic light scattering (DLS) is 56.9 ± 13.6 nm and the polydispersity index (PI) is 0.155. The following figures are the UV-Vis absorption spectra of (**D**) PTE, (**E**) PLGA, and (**F**) PLGA-PTE. Comparison of the visible light absorption spectrum of UV light confirmed that PLGA-PTE was successfully synthesized. (**G**) The cumulative drug release of PTE from PLGA NPs formulation at different time points was detected by UV-Vis spectrophotometer at 310 nm. Triangle symbol indicated the PTE concentration released in water. The data are presented as the mean ± standard deviation of three independent experiments.* *p* < 0.05 compared with 0 h groups.

**Figure 3 ijms-22-02536-f003:**
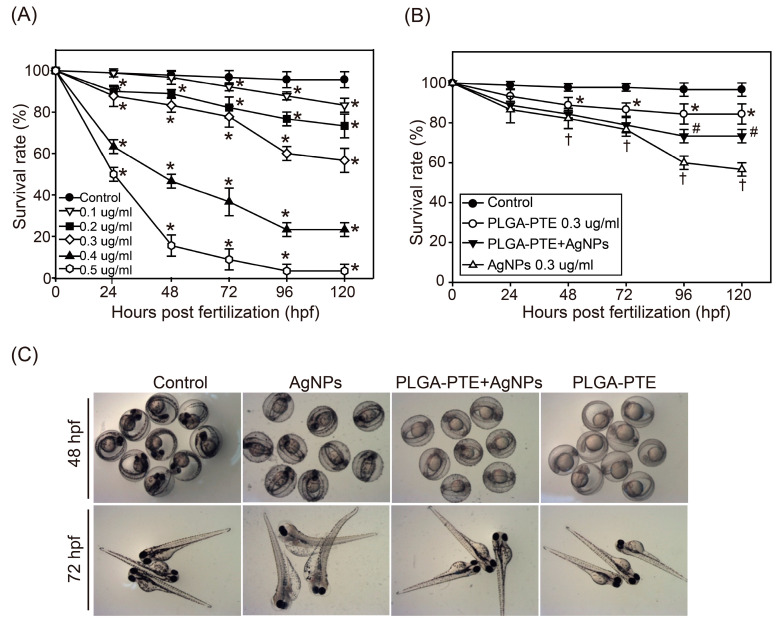
The survival rate of zebrafish embryos. The survival rate of zebrafish embryos exposed to (**A**) AgNPs (0, 0.1, 0.2, 0.3, 0.4, 0.5 μg/mL) for 0, 24, 48, 72, 96, and 120 h post fertilization (hpf) (*n* = 30). The data are presented as the mean ± standard deviation of three independent experiments. (**B**) Survival rate of zebrafish embryos exposed to control, 0.3 μg/mL PLGA-PTE, PLGA-PTE + AgNPs (0.3 μg/mL PLGA-PTE pretreated for 1 h then changed to 0.3 μg/mL AgNPs), or 0.3 μg/mL AgNPs for the indicated time points (*n* = 30). In every treated group, the exposure solutions were replaced every day. The data are presented from at least three independent experiments. * *p* < 0.05, PLGA-PTE versus control groups; # *p* < 0.05, PLGA-PTE + AgNPs versus AgNPs groups; † *p* < 0.05, AgNPs groups versus control groups. (**C**) Appearance of the zebrafish embryos after exposure to control, 0.3 μg/mL AgNPs, PLGA-PTE + AgNPs, and 0.3 μg/mL PLGA-PTE for 48 and 72 hpf under the microscope.

**Figure 4 ijms-22-02536-f004:**
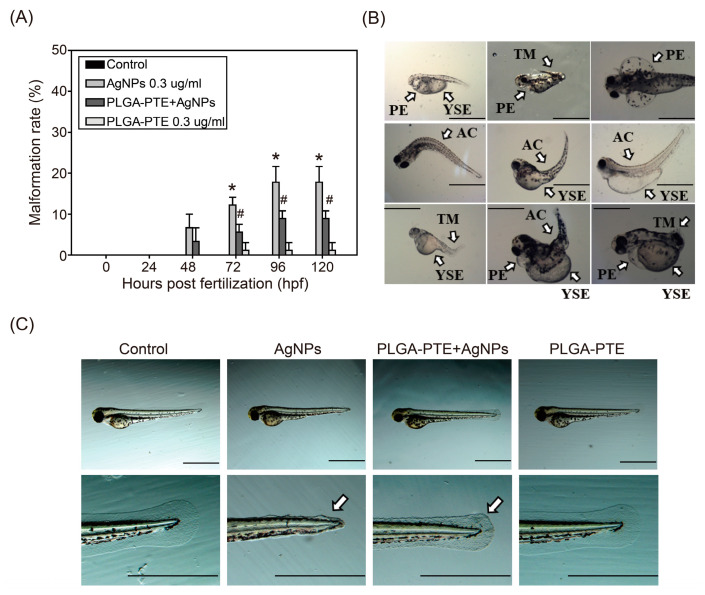
Malformation of zebrafish in response to treatment of AgNPs. (**A**) Malformation percentage of zebrafish embryos after exposure to control, 0.3 μg/mL AgNPs, PLGA-PTE + AgNPs, and 0.3 μg/mL PLGA-PTE for the indicated time points. The data are presented from at least three independent experiments (*n* = 30). * *p* < 0.05, AgNPs groups versus control groups; # *p* < 0.05, PLGA-PTE + AgNPs versus AgNPs groups; (**B**) Malformations of zebrafish embryos caused by AgNPs. PE, pericardial edema; YSE, yolk sac edema; AC, axial curvature; TM, tail malformation. (**C**) Zebrafish embryos after exposure to control, 0.3 μg/mL AgNPs, PLGA-PTE + AgNPs, and 0.3 μg/mL PLGA-PTE for 72 hpf. White arrows showed the malformation of caudal fins. Scale bar = 1.0 mm.

**Figure 5 ijms-22-02536-f005:**
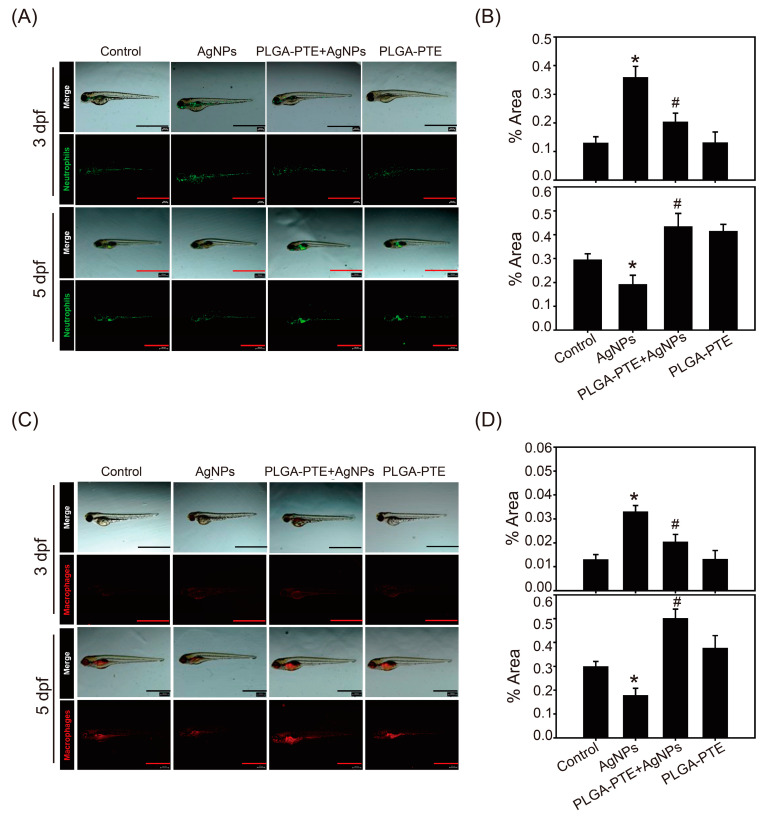
The number and distribution of zebrafish neutrophils and macrophages observed under a stereo fluorescence microscope at 3 dpf and 5 dpf. Zebrafish embryos after exposure to control, 0.3 μg/mL AgNPs, PLGA-PTE + AgNPs, and 0.3 μg/mL PLGA-PTE for 3 and 5 dpf. (**A**) The green fluorescence image shows neutrophils and (**C**) macrophages, which are shown by red fluorescence Scale bar=1.0 mm.. The quantification of fluorescence imaging is shown in (**B**,**D**). The image presented is representative of three independent experiments (*n* = 3). * *p* < 0.05, AgNPs groups versus control groups; # *p* < 0.05, PLGA-PTE + AgNPs versus AgNPs groups.

**Figure 6 ijms-22-02536-f006:**
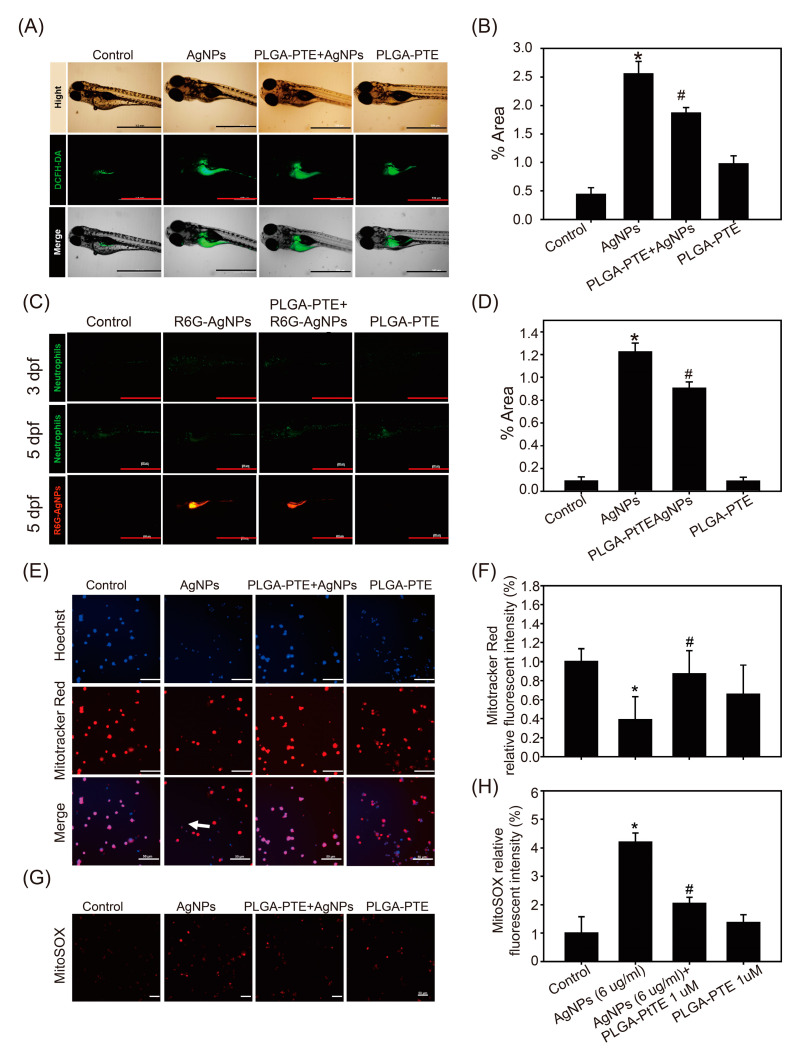
R6G-AgNPs accumulated in zebrafish and affected reactive oxygen species (ROS) production. (**A**) Zebrafish embryos after exposure to control, 0.3 μg/mL AgNPs, PLGA-PTE + AgNPs, and 0.3 μg/mL PLGA-PTE for 120 hpf. Then, ROS formation was shown by staining with DCFH-DA fluorescence dye for 30 min. (**C**) R6G-AgNPs were used to determine the content of AgNPs in zebrafish embryos. Scale bar = 1.0 mm.The fluorescence intensity was quantified in (**B**,**D**). The image presented is representative of three independent experiments (*n* = 3). (**E**) Mitochondrial damage was detected by staining with MitoTracker™ Red (loss of mitochondria membrane potential) and (**G**) MitoSOX™ (indicated ROS production) in THP-1 cells after each treatment for 6 h. The white arrow indicated the cells with damaged mitochondria that was lack of staining with Mitotracker. Scale bar= 50 μm. (**F**,**H**) The image was quantified and presented as the representative of three independent experiments (*n* = 3). * *p* < 0.05, AgNPs groups versus control groups; # *p* < 0.05, PLGA-PTE + AgNPs versus AgNPs groups.

**Figure 7 ijms-22-02536-f007:**
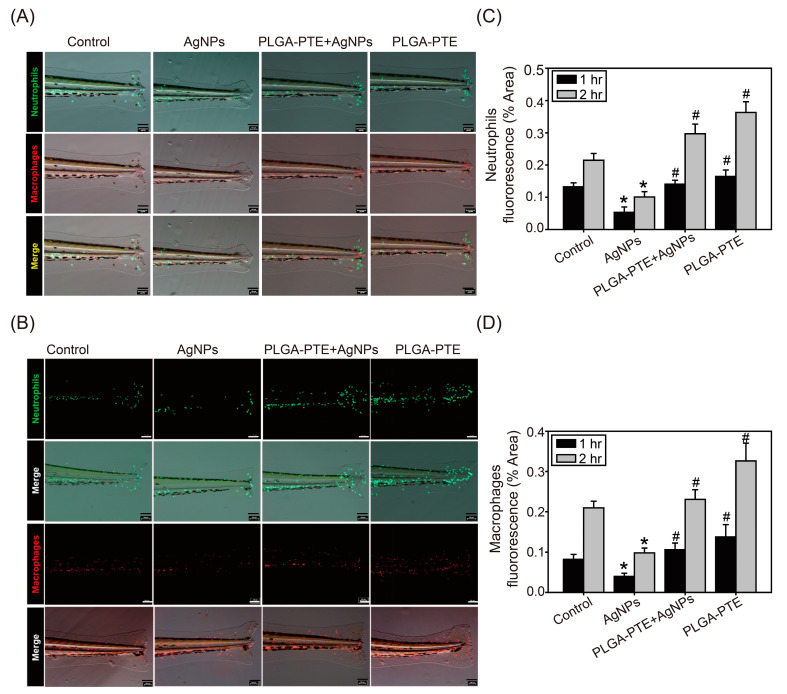
AgNPs exposure affected the accumulation of neutrophils and macrophages in the wounded tail. Zebrafish embryos after exposure to control, 0.3 μg/mL AgNPs, PLGA-PTE + AgNPs, and 0.3 μg/mL PLGA-PTE for 120 h. After the transgenic zebrafish Tg(mpx:eGFP):Neutrophils and Tg(mpeg1:mCherry):Macrophages were anesthetized, their tail fins were transected at the end of the spinal cord with a sterile blade under a microscope. The reverse migration of the immune cells was observed under a fluorescence microscope after (**A**) 1 h and (**B**) 2 h of transection. Scale bar= 100 μm. The image was quantified as (**C**) and (**D**) and the presented is representative of three independent experiments (*n* = 3). * *p* < 0.05, AgNPs groups versus control groups; # *p* < 0.05, PLGA-PTE + AgNPs versus AgNPs groups.

**Figure 8 ijms-22-02536-f008:**
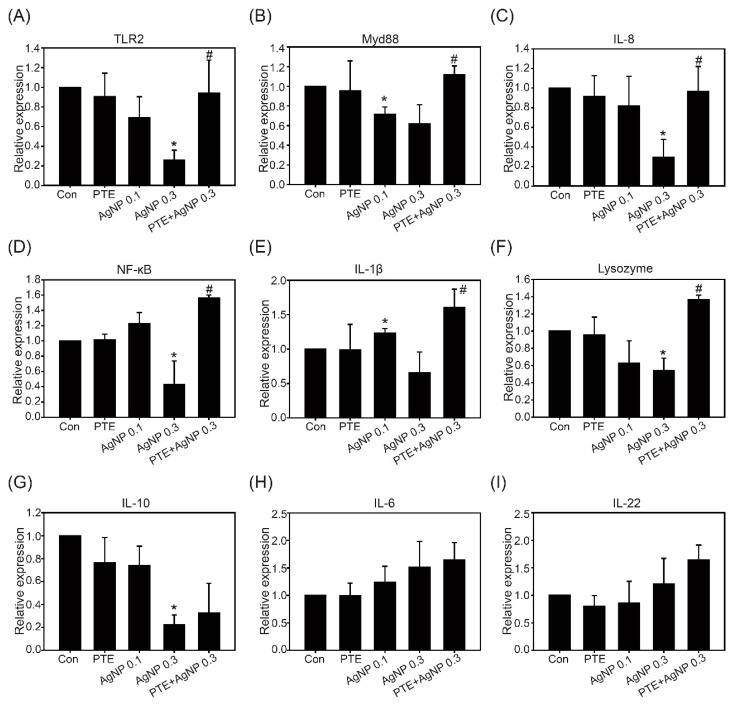
Expression of immune-related genes after 5 dpf exposure. Zebrafish embryos after exposure to control, 0.3 μg/mL PLGA-PTE, 0.1 μg/mL AgNPs, 0.3 μg/mL AgNPs, and PLGA-PTE + AgNPs for 120 hpf. Then, the genes from whole larvae, including: (**A**) Toll-like receptors 2 (*TLR2*), (**B**) *Myd88*, (**C**) *IL-8*, (**D**) *NF-κB*, (**E**) *IL-1β*, (**F**) Lysozyme, (**G**) *IL-10*, (**H**) *IL-6*, and (**I**) *IL-22*, were analyzed. β-actin served as the internal control. The data are presented from at least three independent experiments. * *p* < 0.05, AgNPs groups versus control groups; # *p* < 0.05, PLGA-PTE + AgNPs versus AgNPs groups.

**Table 1 ijms-22-02536-t001:** Characterization of AgNPs.

Characterization	Method	Condition	Results
Size	TEM	Dry	8.89 ± 1.68 nm
Morphology	TEM	Dry	Spherical
Composition and purity	EDX	Dry	Ag (99.6%)
Zeta potential	PALS	Water	−28.85 mV
Hydrodynamic size	DLS	Water	13.2 ± 3.1 nm
Polydispersity index (PDI)	DLS	Water	1.32 × 10^−8^
λ_max_	UV-Vis	Water	391 nm

**Table 2 ijms-22-02536-t002:** Primers for RT-qPCR.

Gene Name	5′ → 3′ Forward Sequence	3′ → 5′ Reverse Sequence
*IL-1β*	TTGAAAGTGCGCTTCAGCA	CGGTCCTTCCTGAAGAACA
*IL-6*	AGACCGCTGCCTGTCTAAAA	TTTGATGTCGTTCACCAGGA
*IL-8*	GTCGCTGCATTGAAACAGAA	CTTAACCCATGGAGCAGAGG
*IL-10*	TCACGTCATGAACGAGATCC	CCTCTTGCATTTCACCATATCC
*IL-22*	CATCGAGGAACAACGGTGTACA	CACGAGCACAGCAAAGCAAT
*NF-κB*	AGGAGCGCAGGATACACAG	CAGGAAACAGCTTCTCCCAC
*TLR2*	TGTCTCCCACCCTGAAACTC	TAGTGCCACCTTCCTTCACC
*Myd88*	TCCGAAAGAAACTGGGTCTG	TCGTCATCTAAAATTTCTTTGAGC
Lysozyme	GATTTGAGGGATTCTCCATTGG	CCGTAGTCCTTCCCCGTATCA

## Data Availability

Not applicable.

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
