# Peer review of "Modulation of Innate Immune Toxicity by Silver Nanoparticle Exposure and the Preventive Effects of Pterostilbene"

_ijms, 2021, doi:10.3390/ijms22052536_

Round 1

Reviewer 1 Report

The article “Modulation of Innate Immune Toxicity by Silver Nanoparticle 2 Exposure and the Preventive Effects of Pterostilbene” by Dr Chen et al. reports the effect of pterostilbene conjugated with PLGA to contrast the toxicity of silver nanoparticles on zebrafish models.

The article as whole is well organized but it may be considered after addressing issues:

  • The acronym used in the text to indicate the pterostilbene could be confused with platinum, it would be better to use PTE.
  • Figure 2 and figure 3 are confused. Has figure 3 been omitted?
  • The authors mention HPLC analysis but the methods have not been described adequately and the chromatograms have been omitted. Also UV spectra have to be attached.
  • For a better completeness of the article could be interesting to evaluate the mitochondrial damage.
  • What does the data analysis at the end of the manuscript refer to?
  • Conclusion have to be added.

Author Response

Reviewer 1:

The article “Modulation of Innate Immune Toxicity by Silver Nanoparticle 2 Exposure and the Preventive Effects of Pterostilbene” by Dr Chen et al. reports the effect of pterostilbene conjugated with PLGA to contrast the toxicity of silver nanoparticles on zebrafish models.

The article as whole is well organized but it may be considered after addressing issues:

Comment:

  1. The acronym used in the text to indicate the pterostilbene could be confused with platinum, it would be better to use PTE. Figure 2 and Figure 3 are confused. Has Figure 3 been omitted?

Response:

Thank you for the suggestion. We have changed the abbreviation of Pterostibene to PTE as suggested by the reviewer. We have also included the explanation of Figure 3 as shown in the results (line 133 to 138):

“Zebrafish embryos were exposed to AgNPs (0.1, 0.2, 0.3, 0.4, and 0.5 μg/ml) from 0-120 hpf (Fig. 3A). The results showed that AgNPs reduced the survival of the embryos in a dose- and time-dependent manner. In contrast, pretreatment with 0.3 μg/ml PLGA-PTE followed by 0.3 μg/ml AgNPs for 0-120 hpf significantly reversed the mortality of the embryos (Fig. 3B). Fig. 3C further shows that embryos exposed to AgNPs had higher aggregation of NPs in the chorion, whereas PLGA-PTE reduced the aggregation of AgNPs.”

Comment:

  1. The authors mention HPLC analysis but the methods have not been described adequately and the chromatograms have been omitted. Also UV spectra have to be attached.
  2.  

Response:

Thank you for pointing out this issue. We have included the methods of HPLC analysis in line 422 to 424, as well as the chromatograms in Fig. 2A (line 126). The UV spectra ranged from 200 nm to 800 nm, as shown in line 428.

Materials and Methods section:

5.1. Synthesis of AgNPs and PLGA-PTE

The detector used was Waters Nova-Pak C-18 column (150 × 3.9 mm, 5 μm particle size). The mobile phase is a 50:50 vol./vol. mixture that has been filtered and degassed under reduced pressure.

5.2. AgNPs and PLGA-PTE characterization

Following the syntheses, the AgNPs and PLGA-PTE were characterized. For stability testing, an ultraviolet-visible (UV-Vis) spectrophotometer (Thermo Scientific™ NanoDrop 2000c, NanoDrop Technologies, Thermo Fisher Scientific, Inc., Pittsburgh, PA, USA) was used. The UV spectra ranged from 200 to 800 nm.

Comment:

  1. For a better completeness of the article could be interesting to evaluate the mitochondrial damage.

Response:

Thank you for the suggestion. To evaluate the mitochondrial damage, we have performed mitochondrial membrane potential (MMP) analysis and mitochondrial superoxide analysis. The relevant results were shown in Figure 6. The data was included in Fig. 6E~6H. The description of results and analytic methods were added in line 180 to 190 and line 482 to 493, respectively.

Results section:

“Previous studies have indicated that mitochondria is likely to be the intracellular target of AgNPs toxicity [20]. AgNPs induced brain toxicity has been associated with abnormal mitochondrial function and the upregulation of the genes relevant to innate immunity [21]. Therefore, we further explored whether PLGA-PTE restores the function of immune cells by regulating mitochondria damage. Human macrophage cell line THP-1 cells were treated with AgNPs for indicated time point and the results showed that AgNPs significantly reduced mitochondrial membrane potential (MMP) (Fig. 6E and 6F) and induced ROS production (Fig. 6G and 6H) as evidenced by MitoTracker™ Red and MitoSOX™ staining, respectively. Meanwhile, pretreatment with PLGA-PTE for 1 hr prior to AgNPs treatment has significantly protected against MMP loss and ROS production in macrophages. These findings suggested that mitochondria is a target for AgNPs toxicity, while PLGA-PTE protected against mitochondria damage, and thereafter reversed MMP and reduced ROS production. The in vitro results was consistent to the findings observed in zebrafish (Fig. 6).”

Materials and Methods section:

5.7. Mitochondrial damage analysis

To analyze the mitochondrial damage induced by exposure to AgNPs, mitochondrial membrane potential (MMP) analysis and mitochondrial superoxide analysis were performed by using THP-1 cells. For MMP analysis, THP-1 cells were exposed to 1 µM PLGA-PTE for 1 hr prior to exposure to 6 µg/ml AgNPs. After 6 hrs of exposure, the level of MMP was measured using the staining of MitoTracker® Red CMXRos probe (M5712, Invitrogen, Rockford, IL, USA). Cells were treated with 100 nM MitoTracker® Red CMXRos and incubated at 37ºC for 30 min and the nuclei were visualized by Hoechst 33258. Images were obtained with Nikon H600L microscope (Nikon, Japan) and the relative fluorescent intensity was measured by ImageJ software.

For mitochondrial superoxide analysis, THP-1 cells were exposed to 1 µM PLGA-PTE for 1 hr prior to exposure to 6 µg/ml AgNPs. After 1 hr of exposure, the level of mitochondrial superoxide was measured using MitoSOX™ Red probe (MM36008, Invitrogen, Rockford, IL, USA). The images were obtained with Nikon H600L microscope (Nikon, Japan) and the relative fluorescent intensity was measured by ImageJ software.

Comment:

  1. What does the data analysis at the end of the manuscript refer to?
  2.  

Response:

Thank you for the comment. The section refers to statistical analysis performed. To avoid any misunderstanding, we have revised the title of Section 5.9 to “Statistical analysis” as shown in line 504.

Comment:

  1. Conclusion have to be added.
  2.  

Response:

Thank you for the suggestion. We have added the conclusion in Section 4, as shown in line 390 to 407.

Conclusion

“The ubiquity of silver nanoparticles (AgNPs) in various consumer products has raised the concern for its safety. Multiple publications has proven that exposure to AgNPs will induce toxic effects in the human body. In this study, we have shown the effects of AgNPs toxicity on immunological functions by using a zebrafish model. Exposure to AgNPs has induced innate immune toxicity, morphological malformations, as well as death in zebrafish embryos. Interestingly, we have proven that the addition of pterostilbene (PTE), a naturally recurring stilbene compound, could provide protection against the AgNPs-induced immunotoxicity. We found a significant decrease of the toxic effects found in the groups of zebrafish that has been pretreated by PLGA-PTE compared to those without PLGA-PTE pretreatment. Due to the anti-inflammatory and antioxidant properties of PTE, previous publications have stated that PTE was found to inhibit the production of ROS in in vivo arthritis model [39, 43]. Because PTE is a lipophilic compound, we encapsulated PTE with PLGA to increase its water solubility, thereby increasing the rate of PTE consumption by zebrafish. The release rate of PTE within 24 hrs in water is only about 30%, indicating that zebrafish could have consumed enough amount of PLGA-PTE and thereafter induce the protective effects against AgNPs toxicity. Accordingly, our study has proven that the addition of PLGA-PTE can provide a safer application of AgNPs and restore the AgNPs-induced innate immune toxicity. The protective mechanisms of PLGA-PTE in zebrafish immune system could be partially explained by mitochondria protection and reducing ROS production of. Although the detailed protective mechanisms are warant for additional investigation, we hope that this study can initiate future studies regarding how PTE can be used to reduce the toxic effects of AgNPs in various aspects.”

Reviewer 2 Report

This is a well written manuscript by Chen et al. It explores the immune modulation and toxic effects of AgNPs using a Zebra fish model. They also examine if pretreatment with pterostilbne (Pt) will have beneficial effect to circumvent the AgNPs induced toxicity. Their data indicates the effect of AgNPs in Zebra fish model and probable beneficial effect of pretreatment with PLGA-Pt. My comments to improve the manuscript include:

  1. It is not clear why they choose 0.3ug/mL of AgNPs. I assume figure 3 might have the answer for this but figure 3 is missing in the manuscript.
  2. What is the rationale of using 0.3ug/mL of PLGA-Pts? Is that the concentration of the mix or Pts alone? How was the Pts release analyzed?
  3. Result 2.1 shows more than the characterization of AgNPs and PLGA-Pt. So, change this title to reflect the data shown up to figure 4.
  4. Is there possibility of quantifying the neutrophil/macrophage migration/reverse migration?
  5. In discussion section- it is not clear on the translational value of PLGA-Pt or Pt alone. How can this be used in real life scenarios to meet the ultimate goal of the researcher which is to reduce the toxic effects of AgNPs? Obviously, this research does not answer that but the authors should make it clear and explain as the future perspectives.

Author Response

Reviewer 2:

This is a well written manuscript by Chen et al. It explores the immune modulation and toxic effects of AgNPs using a Zebra fish model. They also examine if pretreatment with pterostilbne will have beneficial effect to circumvent the AgNPs induced toxicity. Their data indicates the effect of AgNPs in Zebra fish model and probable beneficial effect of pretreatment with PLGA-Pt. My comments to improve the manuscript include:

Comment:
1.     It is not clear why they choose 0.3ug/mL of AgNPs. I assume figure 3 might have the answer for this but Figure 3 is missing in the manuscript.

Response:

Thank you for pointing out this issue. The rationale behind choosing this concentration was based on the LC50 that was found to be nearly 0.3 μg/ml (line 463 to 464). We have also included Figure 3 as shown in line 143.

Materials and Methods section:

Rationale behind choosing 0.3 μg/ml:

“At the end of the 120 hpf experiment, the LC50 was found to be nearly 0.3 μg/ml; hence, this dose was selected for subsequent studies.”

Comment:
2.     What is the rationale of using 0.3ug/mL of PLGA-Pts? Is that the concentration of the mix or Pts alone? How was the Pts release analyzed?

Response:

Thank you for the comment. The rationale behind choosing this concentration was based on the survival rate of the embryos that was found to be 90% after treatment of 0.3 μg/ml PLGA-PTE (line 466 to 467). We have also performed the drug loading and drug release analysis of PLGA-PTE as shown in the “Results”, “Material and Methods”, “Figure 2G”, and “Figure legends”.

Results section:

“In addition, the release rate of PTE from PLGA in water that was detected by UV-Vis spectrophotometer showed that PLGA particles provided a sustained release of PTE of linear character, following an initial burst period (releasing approximately 30% within the first day). The release rate of PLGA-PTE in water is around 30% - 50% within 120 hrs (Fig. 2G).”

Materials and Methods section:

Rationale behind choosing 0.3 μg/ml PLGA-PTE:

“At the end of the 120 hpf experiment, the embryo survival rate at treatment of 0.3 μg/ml PLGA-PTE was found to be 90%; hence, this dose was selected for subsequent studies.”

Drug loading and drug release analysis of PLGA-PTE:

5.3. Drug loading and drug release analysis of PLGA-PTE

For the drug loading study of PLGA-PTE, 500 µl of PLGA-PTE solution was centrfuged at 21,100 RCF for 20 minutes to separate the PLGA-PTE nanoparticles and free PTE. The PLGA-PTE NPs pellet was then redissolved in alcohol and the PTE concentration was analyzed using ultraviolet-visible (UV-Vis) spectrophotometer. The UV spectra ranged from 200 to 800 nm. The maximum absorbance value was detected at 310 nm. The drug loading rate (%) was calculated by using the following formula: (Amount of drug released from the PLGA-PTE/Initial amount of drug for NPs preparation) x 100%. All of the experiments were done in triplicate. The loading rate is around 49.00±1.36 (%).

The release of PLGA-PTE was carried out in distilled water. First, PLGA-PTE was centrifuged at 21,100 RCF for 20 minutes to separate After 6, 24, 48, 72, 96, and 120 hrs, the solution was withdrawn and centrifuged at 21,100 RCF for 20 minutes to separate the nanoparticles and the free PTE. The release of PTE was analyzed with UV-visible spectrophotometry at 310 nm. All of the experiments were done in triplicate.

Comment:
3.     Result 2.1 shows more than the characterization of AgNPs and PLGA-Pt. So, change this title to reflect the data shown up to Figure 4.

Response:

Thank you for the comment. To avoid any misunderstanding, we have arranged the explanation for Figure 1 and Figure 2 in Section 2.1 “Characterization of AgNPs and PLGA-PTE” (line 86 to 100), and the explanation for Figure 3 and Figure 4 in Section 2.2 “Effects of AgNPs and PLGA-PTE on zebrafish embryos” (line 133 to 142).

Comment:
4.     Is there possibility of quantifying the neutrophil/macrophage migration/reverse migration?

Response:

Thank you for the comment. We have added the quantified results in Fig. 7C, 7D, and in the Figure legends.

Figure legends for Fig. 7:

“The image was quantified as (C) and (D) and the presented is representative of three independent experiments (n = 3). *p < 0.05, AgNPs groups versus control groups; #p < 0.05, PLGA-PTE + AgNPs versus AgNPs groups.”

Comment:
5.     In discussion section- it is not clear on the translational value of PLGA-Pt or Pt alone. How can this be used in real life scenarios to meet the ultimate goal of the researcher which is to reduce the toxic effects of AgNPs? Obviously, this research does not answer that but the authors should make it clear and explain as the future perspectives.

Response:

Thank you for the suggestion. We have added a discussion regarding the applications of pterostilbene (line 366 to 369 and line 335 to 340) and its future perspectives in the Section 4 “Conclusion” (line 390 to 407).

Discussion section:

Line 366 to 369:
“As the world keeps searching for alternative natural compounds, the translational values of stilbene compounds are hard to ignore. It has been proven in many publications that PTE, one of the stilbene compounds, is beneficial on various levels and applications. Notably, our study shows that pretreatment with PTE increased the survival rate of embryos exposed to AgNPs and reduced the occurrence of malformations (Figs. 3, 4).”

Line 384 to 388:

Previously, we have gathered evidence on how PTE can suppress oxidative stress, inflammation and modulates senescence, apoptosis, autophagy, and cell cycle arrest [14]. In addition, PLGA-capped PTE can increase the solubility of PTE in water (Fig. 2) and change the characterization of PTE to provide better bioavailability in zebrafish. However, further investigation of how PLGA-PTE protected against innate immune toxicity after nanoparticle exposure is still needed. In summary, the addition of PTE is a promising solution to ameliorate the immune toxicity induced by AgNPs, thereby providing a safer approach for its application.

Conclusion section:

“The ubiquity of silver nanoparticles (AgNPs) in various consumer products has raised the concern for its safety. Multiple publications has proven that exposure to AgNPs will induce toxic effects in the human body. In this study, we have shown the effects of AgNPs toxicity on immunological functions by using a zebrafish model. Exposure to AgNPs has induced innate immune toxicity, morphological malformations, as well as death in zebrafish embryos. Interestingly, we have proven that the addition of pterostilbene (PTE), a naturally recurring stilbene compound, could provide protection against the AgNPs-induced immunotoxicity. We found a significant decrease of the toxic effects found in the groups of zebrafish that has been pretreated by PLGA-PTE compared to those without PLGA-PTE pretreatment. Due to the anti-inflammatory and antioxidant properties of PTE, previous publications have stated that PTE was found to inhibit the production of ROS in in vivo arthritis model [39, 43]. Because PTE is a lipophilic compound, we encapsulated PTE with PLGA to increase its water solubility, thereby increasing the rate of PTE consumption by zebrafish. The release rate of PTE within 24 hrs in water is only about 30%, indicating that zebrafish could have consumed enough amount of PLGA-PTE and thereafter induce the protective effects against AgNPs toxicity. Accordingly, our study has proven that the addition of PLGA-PTE can provide a safer application of AgNPs and restore the AgNPs-induced innate immune toxicity. The protective mechanisms of PLGA-PTE in zebrafish immune system could be partially explained by mitochondria protection and reducing ROS production of. Although the detailed protective mechanisms are warant for additional investigation, we hope that this study can initiate future studies regarding how PTE can be used to reduce the toxic effects of AgNPs in various aspects.”